# The Green Versus Green Trap and a Way Forward

**Haris Doukas [1],\*, Alexandros Nikas [1] , Giorgos Stamtsis [2] and Ioannis Tsipouridis [3]**

[1]   Energy Policy Unit, School of Electrical and Computer Engineering, National Technical University of Athens, 157 73 Athens, Greece; anikas@epu.ntua.gr

[2]   Hellenic Association of Independent Power Companies, Greece; stamtsis@gmail.com

[3]   Technical University of Mombasa, Mombasa, Coast Province 80100, Kenya; i.tsip@redpro.gr

\*   Correspondence: h_doukas@epu.ntua.gr

**Abstract:** Massive deployment of renewables is considered as a decisive step in most countries' climate efforts. However, at the local scale, it is also perceived by many as a threat to their rich and diverse natural environment. With this perspective, we argue that this green versus green pseudo-dilemma highlights how crucial a broad societal buy-in is. New, transparent, participatory processes and mechanisms that are oriented toward social licensing can now be employed. A novel, integrative research agenda must orbit around co-creation to enable and promote resource co-management and co-ownership where possible, with increased consensus.

**Keywords:** climate; biodiversity; integrative policy support; social participation; renewable energy

## 1. Introduction

The emblematic Intergovernmental Panel on Climate Change (IPCC) 1.5 °C Special Report spelled out that about a decade and half degrees Celsius stand between us and a milestone temperature rise impacting nature and humans alike to a markedly different extent [1]. With almost 80% of all final energy consumption being fuelled by fossils [2], the planet seems locked onto a trajectory leading to a manifold temperature rise by the end of this century.

The majority of scenarios modelled point inter alia to huge investments in renewable energy sources, implying significant land requirements for their development in the near future. At the same time, energy crop cultivation for $CO_2$ absorption to limit warming to 1.5 °C could occupy up to seven million sq. km of land (slightly less than the size of Australia), with uncertain environmental and social trade-offs [3].

Moreover, the global wildlife population has fallen by 60% over the last 40 years, with one million species being at risk of extinction, thereby highlighting the need to increase the number and size of protected areas. This was also reflected in the EU Biodiversity Strategy for 2030 (COM/2020/380), which provides for a larger EU-wide network of protected areas, corresponding to at least 30% of land and sea in Europe, building upon the existing Natura 2000 network, with strict protection for areas of very high biodiversity and climate value.

Additionally, although biodiversity loss and climate change are interdependent and further exacerbate one another, in practice respective mitigating efforts are usually perceived as conflicting shades of green. The most prominent example is the case of renewable energy investments for climate action, which are very frequently unwelcome by a growing number of local communities, on environmental grounds—renewables, and especially wind turbines, are claimed to cause irreversible damage to the environment and land exploitation.

## 2. Evidence from PARIS REINFORCE

Within PARIS REINFORCE [4], a modelling research project envisioned to embody the Talanoa dialogue process in climate science, we started involving a large and diverse pool of stakeholders in all stages of climate policy decision-making. The process began with 57 European policymakers, academics, industry and Non-Governmental Organisation (NGO) representatives in a workshop held in Brussels, on 21 November 2019, and hundreds of others attending the workshop online. The spatial planning of renewables was considered a key research topic, while local communities' sustainability and societal acceptance as key factors for wide-scale deployment of renewables.

A similar workshop of national focus, held at the Acropolis museum in Athens, Greece, on 28 January 2020, was physically attended by 398 individuals; similar findings were reached in terms of topic and risk prioritisation, despite the synthesis gap between the two audiences, with civil society dominating the pool of attendants this time. The prospect of exploiting Greece's abundant yet largely underexploited [5] renewable energy potential faced heated opposition and the most significant outcome of the workshop was reflected in a vivid "Green versus Green" debate. Many participants argued against renewable energy resources (especially wind) projects, mostly from a biodiversity perspective—across all interactive workshop sessions, citizens and environmentalists alike referred to wind-related green grabbing [6] and, most importantly, to green energy projects impinging on environmentally protected areas.

This discussion is particularly important for Greece, where lignite has fuelled electricity generation since the early 1950s and been a major socioeconomic growth driver for decades since. In 2009, lignite-fired generation covered 58% of the electricity demand in the country's mainland interconnected system, while renewables covered only 5%. A decade later, lignite covers a modest 20%, with almost the same amount supplied by renewables, while the government aims to decommission the entire lignite plant fleet by 2023.

Currently, more than 28% of the country's terrestrial area is part of the Natura 2000 network [7], a figure surging to 35.2% [8] if nationally designated areas are also considered. Expectedly, as much as 22.3% of installed wind capacity and 26% of upcoming projects are estimated to fall within the Natura 2000 network alone (see also Figure 1, based on the European Environment Agency, 2020 [9]; the Hellenic Ministry of Environment and Energy, 2020 [10]; the Regulatory Authority for Greece [11]; own elaboration).

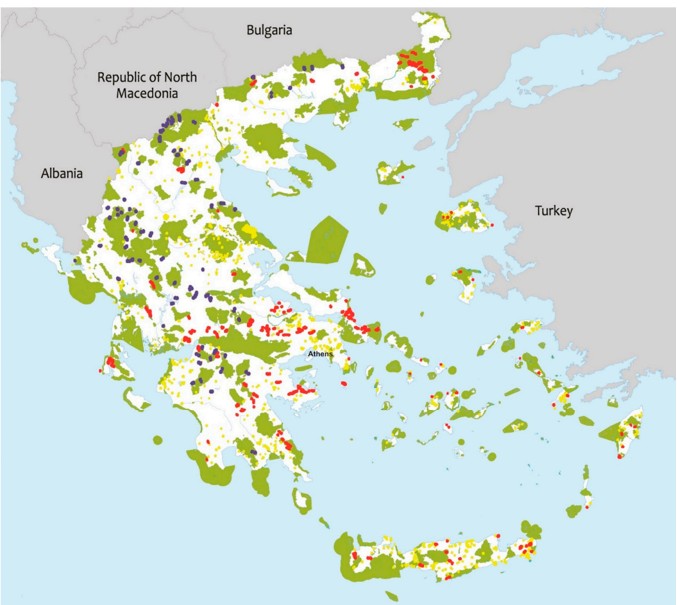

**Figure 1.** Map of Greece, featuring the Natura 2000 network, as well as existing renewable energy projects. Green: protected areas under the European Commission Habitats Directive and/or the

EU Directive on the Conservation of Wild Birds; red: wind power plants; yellow: solar photovoltaic plants; blue: small hydro plants.

## 3. The International Scene

But Greece is not a unique case: the struggle to transition from coal-fired power plants to renewables in many counties highlights the trade-offs between climate change mitigation actions on the one hand, and local environmental protection and biodiversity perceptions and concerns on the other. Upscaling the deployment of renewable energy, despite societal opposition fuelled by citizens' perceptions of environmental degradation or fears of impacts on local environmental, cultural or aesthetic attributes, is a global challenge.

Not-In-My-Backyard (NIMBY) behaviours, partly driven by agendas looking into local wind projects in isolation from global energy transition efforts [12], gradually give rise to an extreme Build Absolutely Nothing Anywhere Near Anything (BANANA) crisis, as evident in many places around the world.

Almost 18% of large-scale renewable energy generation facilities distributed globally are within the boundaries of important conservation areas, and this number could increase by ~42% by 2028 [13]. This is particularly the case for wind and to a lesser extent for solar. Such overlaps of solar or wind energy facilities with conservation areas are mostly found in Europe and Northeast Asia. In India [14], contrary to all other development projects, renewable energy installations do not require an environmental impact assessment, irrespective of the scale, magnitude and area of land required for construction, when situated in or around areas of high biodiversity. In the United States, despite multitudinous studies focusing on links between bird mortality rates and wind turbines, it is biofuels that pose the biggest risk of biodiversity loss, with the number and share of current and projected renewable energy projects in protected areas being relatively low [15].

Germany stands out as the country with the most overlaps between current facilities and important conservation areas, mostly within non-strict protected and key biodiversity areas [16]. In China, finally, most of the overlaps include hydropower plants, while in Spain and Germany they mostly include wind power facilities.

## 4. A New Research Agenda Orienting on Social Licensing

NIMBY is not a new phenomenon, and political backlash against climate and energy policy and infrastructure is a longstanding topic in the academic community. Various systematic studies have conducted research on these themes and identified the drivers of positive or negative attitudes regarding energy technologies [17], including a more encompassing analysis to assess the magnitude and variation in NIMBY behaviour [18,19].

Placing solar at the heart of new investments must be further investigated as a way to avoid disruptions in important conservation areas. High solar irradiation is widely available in low-biodiversity and degraded lands, and there is potential for power to be traded out of such regions. Mutually beneficial relationships between solar plants and ecological systems is of course needed; agrivoltaics are a fine example of an approach to augmenting sustainability across a diverse suite of recipient environments [20].

As proposed for the Brazilian Caatinga, the largest and most diverse dry forest of the Americas, windfarm companies can U-turn into key advocates for promoting environmental protection, by supporting the creation of new protected areas and helping governments sustain high-priority biodiversity conservation areas [21].

Importantly, smaller, dispersed facilities can effectively reduce pressure on ecosystems, as opposed to the centralised model of pharaonic projects [22], which flourished in the fossil fuel era.

To support energy decentralisation, the current research agenda must strive towards new integrative approaches [23], orbiting on co-creation [24] and enabling governance and effective decision-making with

increased consensus. Given the complexity of such transformations, a broad societal buy-in is needed and green design processes must be transparent and open to all citizens [25]. Participatory policies and inclusive citizen engagement have proven essential in many other cases, from global trade [26] to common-pool resources [27], of which renewable energy can be considered one [28].

The role of energy cooperatives is very important in this respect. In Germany, more than 40% of installed renewable capacity belongs to citizens and energy communities, who even have their own distribution networks. In Denmark, which is widely considered a frontrunner in renewable energy penetration in general and offshore in particular, wind farms are only licensed if at least 20% is owned by local communities; in fact, the Middelgrunden offshore wind park, built in 2000, became the first offshore wind project to be based on sale of shares, half-owned by 8500 individual investors and the municipal electricity company. The Netherlands aims for citizens and energy communities to own 50% of the new renewable installed capacity by 2030.

Effectively including local communities in the decision-making and ownership of small, decentralised renewable energy projects should constitute a new, inclusive agenda. When implemented in consideration of viability, technical feasibility, and environmental standards, this agenda can serve as a social licensing protocol that guides the dialogue among local nature conservation teams, associations and communities at an early stage of renewable energy planning, supports local actors to unlock financing opportunities, and allows societal supervision and interventions to solve conflicts. A global body of all the actors' representatives can be set up in this respect, to oversee the implementation of this protocol and intervene via its local and national representatives to solve problems.

## 5. Concluding Remarks

In this perspective, we have drawn from evidence from the PARIS REINFORCE project, with the national context of Greece as a starting point, as well as from the international literature and scene, and discussed instances of green versus green perceptions and of existing mitigation-biodiversity trade-offs fuelling such perceptions. We have also discussed a research agenda oriented toward social licensing and active social participation as a way forward. Our perspective focuses on the lack of societal buy-in on environmental grounds and their perceptions among society, hindering renewable energy expansion and climate action. It, therefore, comes with a caveat: it focuses on the societal desirability of transitions [29], rather than resistance to change from the energy oligopoly and respective hurdles instead of motives for small investors and local actors, as evident for example in the UK [30], Germany [31], Greece [32], and Spain [33], which renders abandoning pharaonic clean energy projects harder.

Science has shown that green versus green is a false dilemma. It has established that there cannot exist sustainable climate action without nature conservation [34]. It has also highlighted the importance of understanding the motives and strategies [35] as well as risk perceptions [36] of actors on the ground.

It is time for the research agenda to move from broadly treating sustainable development goals as trade-offs resulting from mitigation pathways, to integrating modelling exercises in the context of sustainable development, formulating an integrative paradigm. One that promotes energy democracy and desirability in practice by moving beyond the patronising stakeholder engagement model of explaining to citizens what must be done, to helping design a low-carbon agenda where none are left underrepresented, truly reflecting co-creation within energy transition pathways, along with co-ownership whenever feasible.

**Author Contributions:** Conceptualization, A.N., and H.D.; methodology, H.D., A.N., and G.S.; validation, A.N., G.S. and I.T.; formal analysis, H.D., A.N., and G.S.; data curation, A.N., and G.S.; writing—original draft preparation, H.D., A.N.; writing—review and editing, H.D., A.N., G.S., and I.T.; visualization, A.N.; supervision, H.D.; project administration, H.D. All authors have read and agreed to the published version of the manuscript.

**Funding:** This research was funded by European Commission Horizon 2020 Framework Programme, "PARIS REINFORCE" Research and Innovation Project, grant number 820846. The APC was funded by the National Technical University of Athens.

**Acknowledgments:** The most important part of this research is based on the H2020 European Commission Project "PARIS REINFORCE" under grant agreement No. 820846. The sole responsibility for the content of this paper lies with the authors. The paper does not necessarily reflect the opinion of the European Commission. The authors would like to thank Christos Petkidis for his copyediting work.

**Conflicts of Interest:** The authors declare no conflict of interest. The funders had no role in the design of the study; in the collection, analyses, or interpretation of data; in the writing of the manuscript, or in the decision to publish the results.

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
