# Peer review of "The Green Versus Green Trap and a Way Forward"

_energies, doi:10.3390/en13205473_

Round 1
Reviewer 1 Report
The article presents a new, interesting view on the problems related to the acceptance of pro-environmental projects by local communities, especially in areas with a rich biodiversity. The Authors' arguments are logical and well documented. Literature sources are up-to-date and adequate to the issues discussed in the paper.
I only have two minor comments:
- I wonder if all keywords are adequate to the content of the work. It seems to me that ‘social participation’ would be a better term instead of ‘local action’. Maybe it should be also resigned from ‘ownership’, and add ‘renewable energy (projects)’. These keywords that I suggest appear most often in the article.
- Technical note: there are no explanations for footnotes (Lines: 43, 67, 69, 70).
The paper is recommended for publication.
Reviewer 2 Report
I think that this piece of updated perspective on the green vs. green debate is opportune and interesting for the community of readers of this journal.
The project they reflect on in this paper deals with a very rellevant topic and even when short (for the type of submission) the brief paper allows to gain insight into the question on how to decentralize renewables in order to reduce the impact on the environment while also contributing to improve it.
There is however on aspect I miss in this text and this is the issue of the resistance to change from the energy oligopoly in many countries such in the UK (Geels, 2014) Germany (Cherp et al. 2017) or Spain (Gabaldón-Estevan, et al. 2018) that makes it more difficult to abandon the pharaonic projects you mention.
For instance, and for the case of Spain (Gabaldón-Estevan, et al. 2018) characterised the resistance and resilience of fossil fuels regimes and how they forced increased hurdles for small investors in PV in the opposite direction of the good examples you cite (lines 128-135).
I think your paper will gain from incorporating this perspective.
Finally, I think that in line 128 there should be a "The" at the begining of the first sentence (i.e. "The role").
Suggested references:
Cherp, A.; Vinichenko, V.; Jewell, J.; Suzuki, M.; Antal, M. Comparing electricity transitions: A historical analysis of nuclear, wind and solar power in Germany and Japan. Energy Policy 2017, 101, 612–628.
Gabaldón-Estevan, D., Peñalvo-López, E., & Alfonso Solar, D. (2018). The Spanish Turn against Renewable Energy Development. Sustainability, 10(4), 1208.
Geels, F.W. Regime resistance against low-carbon transitions: Introducing politics and power into the multi-level perspective. Theory Cult. Soc. 2014, 31, 21–40.
